# The green algae $CO_2$ concentrating mechanism and photorespiration jointly operate during acclimation to low $CO_2$

Ousmane Dao [1], Marie Bertrand[1], Saleh Alseekh [2,3], Florian Veillet[1], Pascaline Auroy [1], Phuong-Chi Nguyen[1], Bertrand Légeret[1], Virginie Epting[1], Amélie Morin[1], Stephan Cuiné[1], Caroline L. Monteil [1], Luke C. M. Mackinder [4], Adrien Burlacot [5,6], Anja Krieger-Liszkay [7], Andreas P. M. Weber[8], Alisdair R. Fernie [2,3], Gilles Peltier [1] ✉ & Yonghua Li-Beisson [1] ✉

Due to low availability of $CO_2$ in aquatic environment, microalgae have evolved a $CO_2$ concentrating mechanism (CCM). It has long been thought that operation of CCM would suppress photorespiration by increasing the $CO_2$ concentration at the Rubisco active site, but experimental evidence is scarce. To better explore the function of photorespiration in algae, we first characterized a *Chlamydomonas reinhardtii* mutant defected in low-$CO_2$ inducible 20 (LCI20) and show that LCI20 is a chloroplast-envelope glutamate/malate transporter playing a role in photorespiration. By monitoring growth and glycolate excretion in mutants deficient in either CCM or photorespiration, we conclude that: (*i.*) CCM induction does not depend on photorespiration, (*ii.*) glycolate excretion together with glycolate dehydrogenase downregulation prevents the toxic accumulation of non-metabolized photorespiratory metabolites, and (*iii.*) photorespiration is active at low $CO_2$ when the CCM is operational. This work provides a foundation for a better understanding of the carbon cycle in the ocean where significant glycolate concentrations have been found.

Microalgae account for nearly half of the photosynthetic $CO_2$ fixation on Earth[1,2]. They contribute to climate change mitigation by assimilating $CO_2$ and are promising candidates for the production of bio-based compounds[3–5]. Due to the low availability of $CO_2$ in water, most algal cells have developed biophysical $CO_2$ concentrating mechanisms (CCMs), thus increasing the $CO_2$ concentration at the catalytic site of Rubisco and favoring the carboxylase reaction at the expense of the oxygenase. Microalgal biophysical CCMs generally involve carbonic anhydrases located in different cellular compartments, bicarbonate and/or $CO_2$ channels or transporters, and in most cases the pyrenoid (a specific cellular compartment where Rubisco is packed in a liquid phase separated organelle and where $CO_2$ is concentrated)[6–10]. As a consequence of the CCM operation, the production of 2-phosphoglycolate (2-PG) resulting from the oxygenase activity of Rubisco is diminished[11,12]. Because of its inhibitory effect on the Calvin cycle, 2-PG is metabolized and recycled through

[1]Aix-Marseille Université, CEA, CNRS, BIAM, UMR7265, Institut de Biosciences et Biotechnologies Aix-Marseille, CEA Cadarache, Saint-Paul-lez-Durance, France. [2]Department of Molecular Physiology, Max Planck Institute of Molecular Plant Physiology, Potsdam-Golm, Germany. [3]Center of Plant Systems Biology and Biotechnology, Plovdiv, Bulgaria. [4]Centre for Novel Agricultural Products, Department of Biology, University of York, York, UK. [5]Department of Plant Biology, Division of Biosphere Sciences and Engineering, The Carnegie Institution for Science, Stanford, CA, USA. [6]Department of Biology, Stanford University, Stanford, CA, USA. [7]Université Paris-Saclay, Institute for Integrative Biology of the Cell (I2BC), CEA, CNRS, CEDEX, Gif-sur-Yvette, France. [8]Institute of Plant Biochemistry, Cluster of Excellence on Plant Science (CEPLAS), Heinrich Heine University, Düsseldorf, Germany. ✉e-mail: gilles.peltier@cea.fr; yonghua.li@cea.fr

photorespiration. Photorespiration consists of multiple metabolic reactions distributed across different subcellular compartments thus requiring inter-organelle communication[13]. During photorespiration, the 2-PG is converted in the chloroplast into glycolate by the 2-PG phosphatase PGP1. In vascular plants lacking a biophysical CCM, glycolate is converted into glycerate through a series of reactions in peroxisomes and mitochondria. Although photorespiration occurs in both plants and microalgae, its subcellular organization seems to differ. For example, in vascular plants lacking a biophysical CCM, photorespiration requires tight cooperation of chloroplast, peroxisome and mitochondria[13]; whereas in Chlamydomonas, photorespiration depends mainly on cooperation between chloroplast and mitochondria[14,15].

In the model green microalga *Chlamydomonas reinhardtii* (hereafter *Chlamydomonas*), part of the glycolate is transferred to mitochondria and further metabolized into glyoxylate, glycine and serine, the other part is excreted out of the cell[16–18]. During acclimation of *Chlamydomonas* to low CO₂, CCM and photorespiratory genes, which share the same master regulator CIA5, are simultaneously induced[19–21]. Such a feature is intriguing since the activity of CCM supposedly inhibits photorespiration. It was proposed that photorespiration is transiently operational during the acclimation from high to low CO₂ conditions until the CCM is fully induced[7,12], photorespiratory metabolites such as 2-PG possibly acting as signaling molecules triggering CCM induction[12,14,22–24].

Two types of CCM regimes have been described in *Chlamydomonas* depending on external CO₂ levels, the low-CO₂-based CCM (CO₂ air level), in which cells preferentially transport CO₂, and the very-low-CO₂-based CCM, in which specific transporters are induced, such as the ATP-binding cassette transporter High Light Activated 3 (HLA3) involved in active bicarbonate uptake[25,26]. During CCM induction, mitochondria migrate from the chloroplast cup to the periphery of the cell at the vicinity of the plasma membrane[27]. The contribution of mitochondria to the energy supply of the algal CCM has been recently evidenced, and energy trafficking between chloroplast and mitochondria was suggested to supply ATP to CCM bicarbonate transporters[28,29]. Actually, energy exchange between subcellular organelles was evidenced decades ago in microalgae[30–33] and has been reexamined more recently[34–37]. The exchange of reducing power between organelles may operate via "malate shuttles" composed of malate dehydrogenases (MDHs) and membrane transporters of dicarboxylic acids[38–41].

Despite the existence of several genes encoding putative chloroplast malate transporters[41–43], none of them has been characterized in *Chlamydomonas*. Based on the increased expression of Low-CO₂ Inducible 20 (*LCI20*) gene, which encodes a putative malate/2-oxoglutarate transporter, during acclimation to limiting CO₂[19–21], it was proposed that LCI20 may be involved as a malate shuttle in the energy trafficking between chloroplast and mitochondria to feed CCM bicarbonate transporters[44]. By another way, photorespiration may also contribute to the energy supply to external CCM transporters[44,45]. Indeed, during photorespiration NADH is produced during the mitochondrial condensation of two glycine molecules into one serine, the NADH being converted into ATP by the mitochondrial respiratory chain. Until now, the nature of metabolic pathways and the identity of proteins involved in the energy network transferring photosynthetic energy from the chloroplast towards mitochondria to feed the external CCM transporters remain largely unexplored.

In this study, we aimed at better characterizing the relationship between CCM and photorespiration in green algae. For this purpose, we a *Chlamydomonas* mutant deficient in LCI20 and showed that LCI20 is a malate/glutamate transporter involved in photorespiration by supplying amino groups for the mitochondrial conversion of glyoxylate into glycine. By comparing growth properties and glycolate excretion in mutants affected in photorespiration or CCM under various photorespiration regimes, we conclude that photorespiratory metabolites do not contribute to CCM induction, that glycolate excretion avoids toxicity of non-metabolized photorespiratory intermediates and that downstream glycolate metabolism, although occurring when the CCM is functioning under very low CO₂, is not required for CCM operation.

## Results

### *LCI20* encodes a putative chloroplast malate transporter whose expression is modulated by light and CO₂ levels

Three putative chloroplast 2-oxoglutarate/malate transporters (annotated as OMT1, OMT2 and LCI20) orthologues of previously characterized Arabidopsis DiT1 and DiT2 malate transporters were identified in the Chlamydomonas genome[42,43]. Previous transcriptomic analysis performed during a day-night cycle or during limiting CO₂ acclimation showed that the expression of *LCI20* (but not that of *OMT1* and *OMT2*) is strongly induced at the beginning of the light period[46] or after the transition from H-CO₂ (2%) to L-CO₂ (0.04%) or VL-CO₂ (0.01%)[19–21] (Supplementary Fig. 1a, b). A phylogenetic analysis revealed that LCI20, which belongs to the GreenCut2[47], is widespread in *Chlorophyta*, closely related to DiT2 transporters, and evolutionarily divergent from OMT family transporters (Supplementary Figs. 1c and 2). Furthermore, LCI20 has recently been localized in the chloroplast but the enrichment in the envelope was not clearly shown[48,49]. Here, we reexamined the strain expressing *LCI20*-mVenus under the control of *PSAD* promoter and we can see clearly that LCI20 is localized to the chloroplast envelope (Fig. 1a and Supplementary Fig. 3).

### The growth of *lci20* is impaired during a transition from high to very low CO₂

A mutant harboring an insertion in the sixth intron of the *LCI20* locus was obtained from the Chlamydomonas library project (CLiP) collection[50] and we named the mutant here as *lci20* (Fig. 1b). We confirmed the insertion of the paromomycin resistance cassette and the absence of the *LCI20* transcript in *lci20* (Supplementary Fig. 4a, b). We then performed genetic complementation of *lci20* by expressing the full-length genomic *LCI20* coding sequence driven by the PSAD promoter (Supplementary Fig. 4a, b). A LCI20 antibody was produced in this study and used to show the absence of LCI20 in the mutant and its presence in WT and the complemented lines (Fig. 1c).

The growth of *lci20* was then investigated under photoautotrophic conditions in both solid and liquid cultures under various CO₂ levels (Fig. 1d, e; Supplementary Fig. 4c, d). On solid media, *lci20* grew normally under H-CO₂, L-CO₂ or VL-CO₂ when cells were previously acclimated to these conditions (Fig. 1d). However, the growth of *lci20* was severely impaired when transitioning from H-CO₂ to VL-CO₂ (Fig. 1e), with smaller decreases in growth observed after a transition from H-CO₂ to L-CO₂ or from L-CO₂ to VL-CO₂ (Fig. 1e). In liquid media, while *lci20* grew normally in H-CO₂ or during transition from H-CO₂ to L-CO₂, its growth was reduced when expressed on the basis of cell volume during transition from H-CO₂ to VL-CO₂ (Supplementary Fig. 4c, d). Based on these results, we conclude that LCI20 is required for photoautotrophic growth when cells are subjected to a sudden and severe CO₂ limitation.

### *lci20* is affected in photorespiratory glycolate metabolism

To determine whether the mutant phenotype is due to the role of LCI20 in the induction of the CCM or in photorespiration, we performed growth assays on agar plates by transferring H-CO₂ grown cells into VL-CO₂ under photorespiratory (21% O₂) or non-photorespiratory (2% O₂) conditions (Fig. 2a). As a control, we used the CCM1 transcription factor knockout mutant *cia5*, which is defective in the transcriptional induction of CCM and of the photorespiratory pathway[11,45,51,52]. *cia5* grew normally under H-CO₂ but growth was completely abolished when cells were transferred to VL-CO₂,

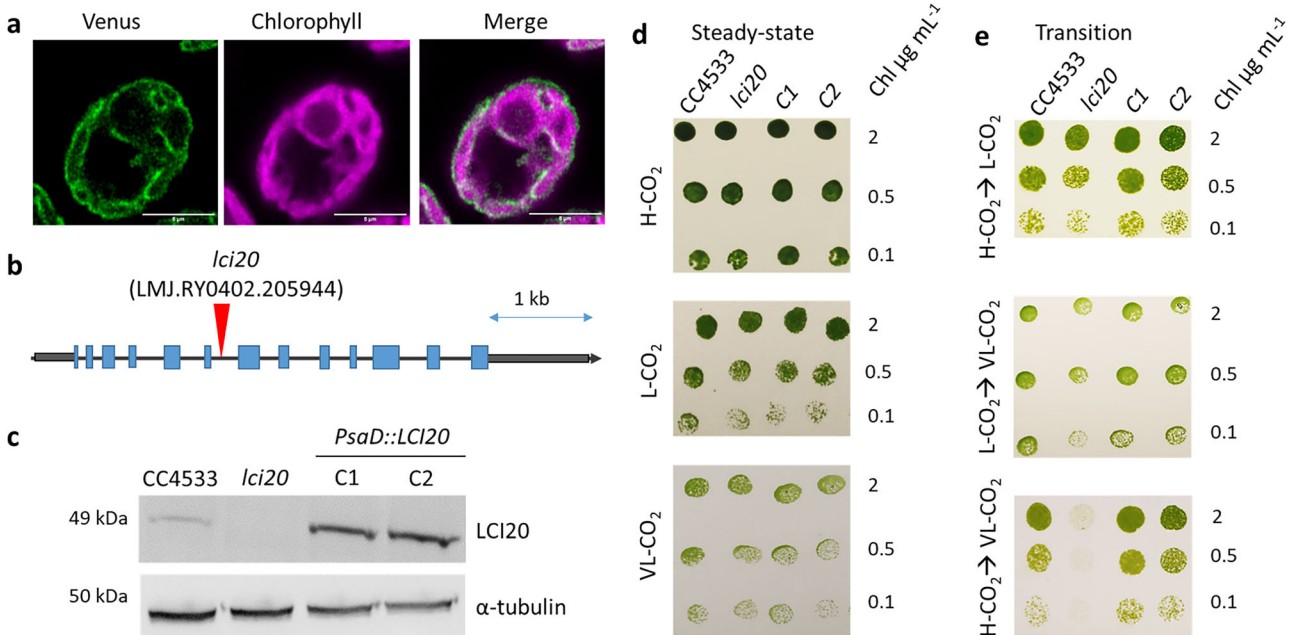

**Fig. 1 | LCI20 subcellular localization, characterization and complementation of a *lci20* insertional mutant. a** Subcellular localization of LCI20 protein fused with the Venus fluorescent reporter. False colours were used to represent Venus (green) and chlorophyll (magenta) fluorescence signals. Scale bar, 5 μm. The localization of LCI20-Venus in the chloroplast envelope was confirmed by three independent experiments. **b** Genomic structure of *LCI20* gene and insertion site of the paromomycin resistance cassette in the CLiP *lci20* mutant. Gray boxes at both extremities represent the 5' and 3'UTR, respectively. Exons are colored blue, and the position of cassette insertion is indicated by red arrows. **c** Immunoblot analysis

using anti-LCI20 antibodies. Uncropped immunoblots are provided as a Source Data file. Due to the limited amount of LCI20 antibodies, this immunoblot was repeated twice with a similar result. Photoautotrophic growth of *lci20*, CC4533 (i.e. WT in this study all throughout) and complemented lines (C1, C2) on agar plates kept under various $CO_2$ regimes. Cells were grown photo-autotrophically in liquid culture under air $CO_2$ level (**d**), or air supplemented with 2% $CO_2$ (**e**) prior to spot testing. Images were taken 3 days (H-$CO_2$ and L-$CO_2$) or 5 days (VL-$CO_2$) after growing under 80 μmol photons $m^{-2}$ $s^{-1}$.

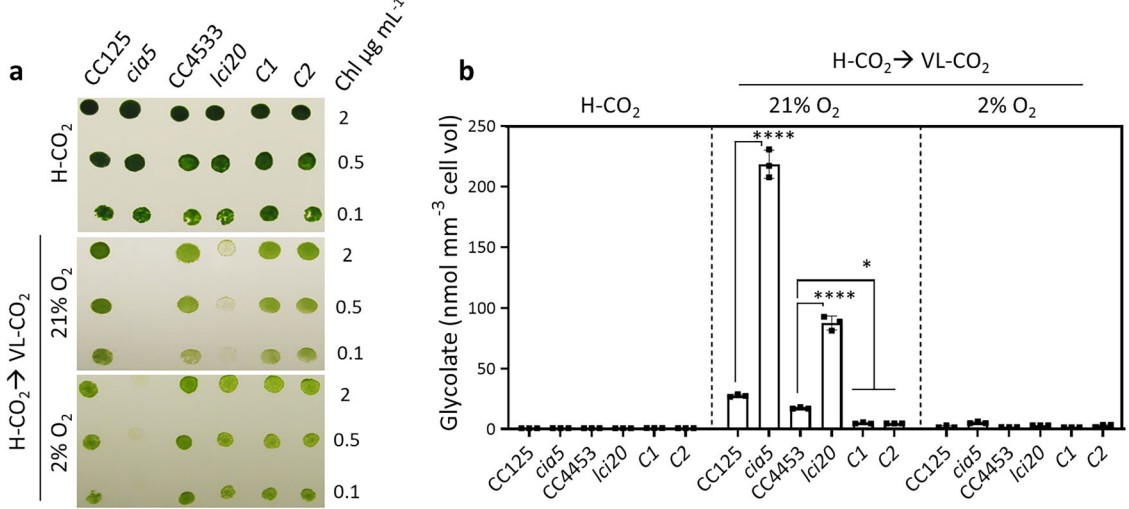

**Fig. 2 | *lci20* is affected in the photorespiratory glycolate metabolism. a** Growth performance of *cia5*, *lci20* mutants and their respective controls when transferring photo-autotrophically grown cells in liquid cultures under H-$CO_2$ to agar plates exposed to VL-$CO_2$ either under 21% or 2% $O_2$ under 80 μmol photons $m^{-2}$ $s^{-1}$. Images were taken after 3 days (H-$CO_2$) or 5 days (VL-$CO_2$) of growth. **b** Quantification of the glycolate concentration in the liquid medium after 20 h

growth of cultures either under H-$CO_2$ or upon transition for H-$CO_2$ to VL-$CO_2$ either in the presence of 21% or 2% $O_2$. Bars show the average and dots show data from independent biological replicates ($n = 3 \pm SD$). Asterisks represent statistically significant difference compared to the wild-type strains ($^*p \leq 0.05$ and $^{****}p \leq 0.0001$) using one-way ANOVA.

irrespective of $O_2$ levels (Fig. 2a), thus indicating that a defect in the CCM, but not in photorespiration, prevented growth of the *cia5* mutant under VL-$CO_2$. Note that similar growth in both solid and liquid cultures was observed when wild-type cells were exposed to VL-$CO_2$ either under photorespiratory conditions (21% $O_2$) or non-photorespiratory conditions (2% $O_2$) (Supplementary Fig. 5), indicating that photorespiration is not required for CCM induction. In contrast, *lci20* grew poorly after transition under photorespiratory conditions (VL-$CO_2$; 21% $O_2$), but the growth was unaffected under non-photorespiratory conditions (VL-$CO_2$; 2% $O_2$) (Fig. 2a), indicating that

*lci20* is defective in photorespiration rather than in CCM. To further distinguish phenotypes between photorespiration and CCM deficiency, we have analyzed the growth of two CCM mutants *bsti* and *cah3* deficient in bestrophins (BST1-3)[53] and lumenal carbonic anhydrase CAH3[54], respectively (Supplementary Fig. 6). Note that *BST1-3* and *CAH3* were reported as low $CO_2$ inducible genes in the same transcriptomic study that has also reported *LCI20* expression[19,21]. In contrast to *lci20* but similar to *cia5* (Fig. 2a), both *bsti* and *cah3* were unable to grow under VLC regardless of the photorespiratory status (Supplementary Fig. 6a), indicating a defect in CCM but not in photorespiration.

Glycolate is excreted into the culture medium by *Chlamydomonas* cells when either the CCM or the glycolate metabolism is defective[16,18,45,55,56]. Glycolate excretion was measured after 20 h acclimation of $H$-$CO_2$ grown cells to VL-$CO_2$ either at 21% or 2% $O_2$. At 21% $O_2$, both *cia5* and *lci20* mutants showed a significantly higher excretion compared to their respective controls (Fig. 2b). Similarly, *bsti* and *cah3* showed higher and continuous glycolate excretion during acclimation

to VL-$CO_2$ at 21% $O_2$ (Supplementary Fig. 6b). Note that we did not detect glycolate excretion under $H$-$CO_2$ or VL-$CO_2$ at 2% $O_2$, conditions when photorespiration is strongly reduced (Fig. 2b). We conclude that *lci20* is defective in the photorespiratory glycolate metabolism, which is triggered when *Chlamydomonas* cells are transferred from $H$-$CO_2$ to VL-$CO_2$ at 21% $O_2$.

## Changes in abundance of photorespiration and CCM proteins during cells' response to different $CO_2$ and $O_2$ levels

To identify possible causes for the growth defect and high glycolate excretion observed in *lci20*, we probed the abundance of different photorespiratory enzymes including the glycolate dehydrogenase (GYD1), hydroxypyruvate reductase (HPR1) and glycine cleavage system P (GCSP) proteins together with CCM-related proteins HLA3, low $CO_2$-inducible 1 (LCI1) and low $CO_2$-inducible C (LCIC) following a transfer from $H$-$CO_2$ to VL-$CO_2$ either at 21% or at 2% $O_2$ (Fig. 3a and Supplementary Fig. 7). We observed a lower abundance of GYD1 in *lci20* compared to control strains when the transfer was performed

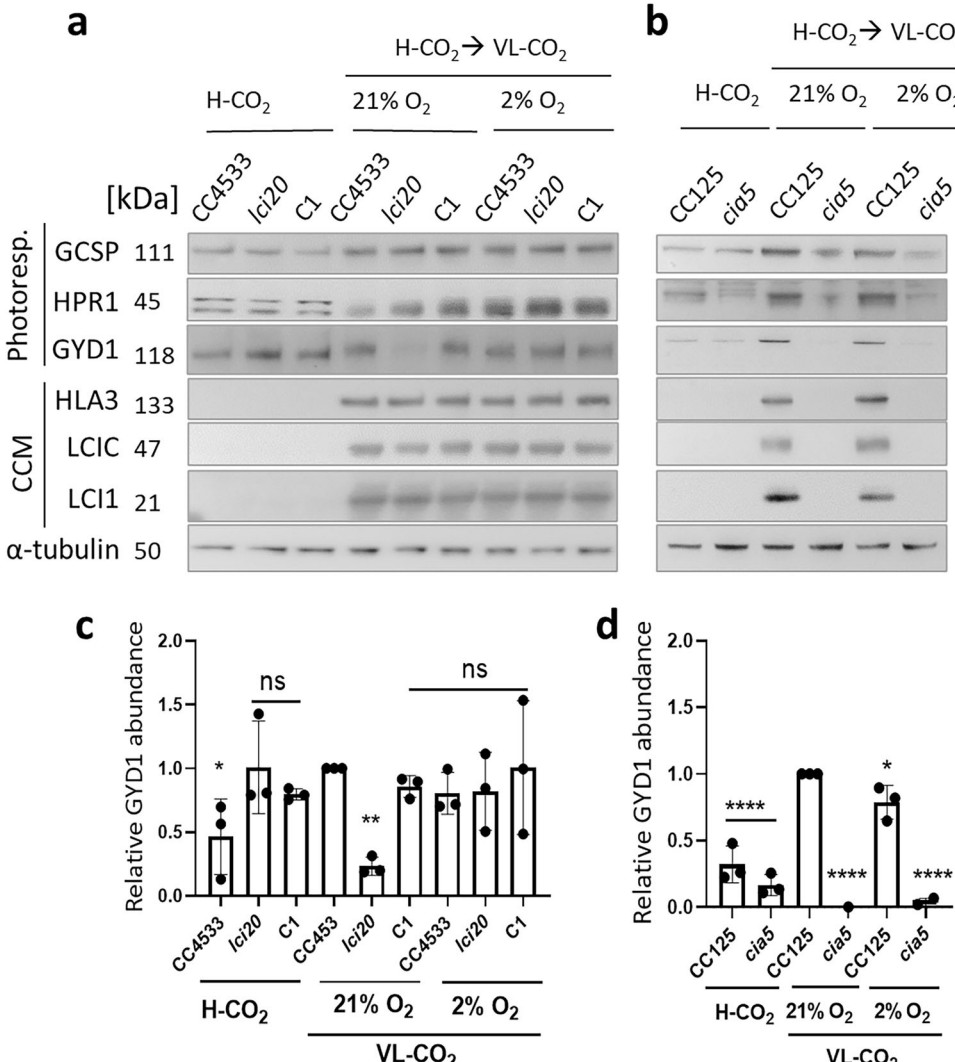

**Fig. 3 | Effect of $CO_2$ and $O_2$ concentrations on the abundance of photorespiration and CCM proteins. a** Immunoblot analysis of representative CCM and photorespiration related proteins in *lci20*, its wild-type control CC4533 and one complemented line (C1). **b** Immunoblot analysis of representative CCM and photorespiration related proteins in *cia5* and its wild-type control CC125. α-tubulin was used as a loading control. Numbers after protein names represent the calculated molecular weights. Relative abundance of GYD1 protein in *lci20* (**c**) and *cia5* (**d**) grown under various $CO_2$ and $O_2$ levels. The GYD1 immunoblots were normalized by α-tubulin signals. Data are means of three independent biological replicates ($n = 3 \pm SD$). ns, not significant. Asterisks represent statistically significant difference compared to the wild-type CC4533 and/or CC125 under VL-$CO_2$ 21% $O_2$ (*$p \le 0.05$, **$p \le 0.01$, and ****$p \le 0.0001$) using one-way ANOVA. Cells were cultivated photo-autotrophically under $H$-$CO_2$ and 80 µmol photons $m^{-2}$ $s^{-1}$ and then acclimated for 20 h at the indicated $CO_2$ and $O_2$ levels prior to immunoblot analysis. Uncropped immunoblots are provided as a Source Data file.

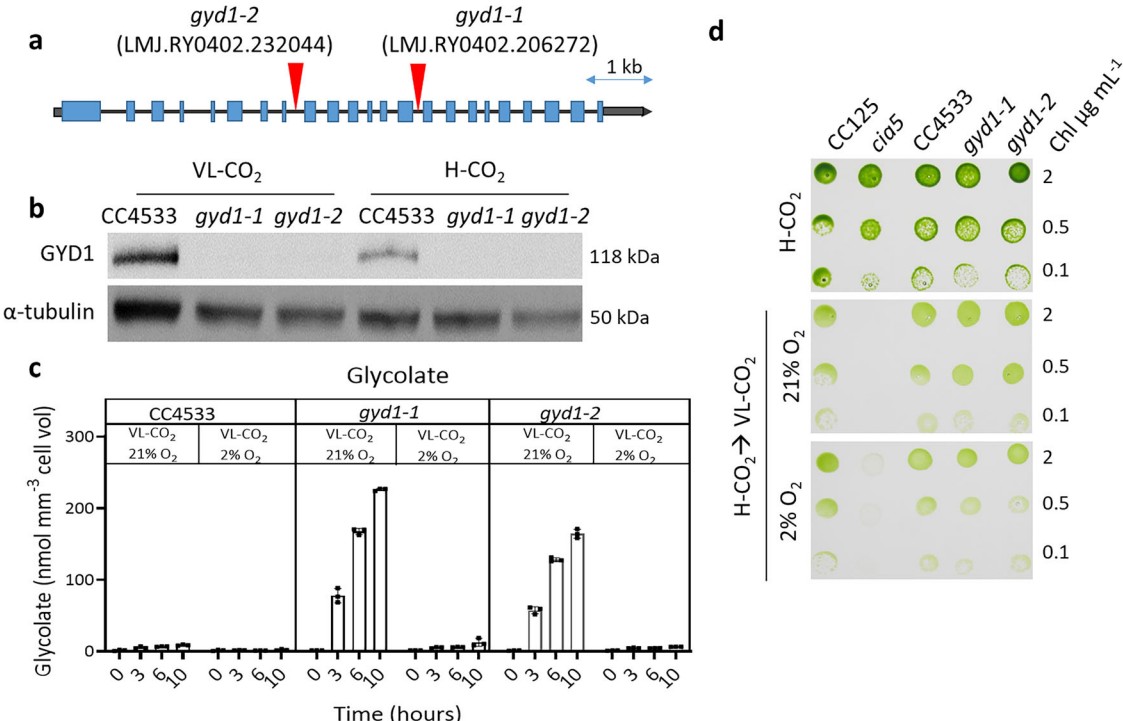

**Fig. 4 | Characterization of two insertional mutants of the glycolate dehydrogenase gene *GYD1*. a** Genomic structure of *GYD1* gene and insertion sites of the paromomycin resistance cassettes in the two CLiP mutants *gyd1-1* and *gyd1-2*. Gray boxes at both extremities represent the 5' and 3'UTR, respectively. Exons are colored blue and the position of cassette insertion is indicated by red arrows. **b** Immunoblot analysis using anti-GYD1 antibodies performed on both *gyd1* mutants and their WT control grown under H-CO$_2$ or VL-CO$_2$. α-tubulin was used as a loading control. Uncropped immunoblots are provided as a Source Data file.

**c** Quantification of the glycolate concentration in the culture medium after a transition from H-CO$_2$ to VL-CO$_2$ at 21% or 2% O$_2$. Bars show the average and dots show data from independent biological replicates ($n = 3 \pm$ SD). **d** Photoautotrophic growth of both *gyd1* mutants, their wild-type control CC4533 and *cia5* and its wild-type control CC125 on agar plates exposed to various CO$_2$ and O$_2$ concentrations. Cells were grown in liquid culture in flasks photo-autotrophically under H-CO$_2$ prior to spot test. Images were taken after 3 days (H-CO$_2$) or 5 days (VL-CO$_2$) of growth under 80 µmol photons m$^{-2}$ s$^{-1}$.

under photorespiratory conditions (21% O$_2$), but no difference was observed under non-photorespiratory conditions (2% O$_2$) (Fig. 3a, c) indicating that the down-regulation of GYD1 in *lci20* is caused by the activity of photorespiration. The abundances of other photorespiration or CCM proteins were not consistently different in all three biological replicates between *lci20* and control strains (Fig. 3a and Supplementary Fig. 7).

In *Chlamydomonas*, photorespiration and CCM genes are co-regulated during acclimation to sub-optimal CO$_2$ and controlled by a common transcription factor CIA5. We then monitored the accumulation of photorespiratory enzymes (GYD1, HPR1 and GCSP) and of CCM-related proteins (HLA3, LCI1 and LCIC) following acclimation to different CO$_2$ and O$_2$ levels in *cia5* and its wild-type control (Fig. 3b and Supplementary Fig. 8). Whereas CCM and photorespiratory proteins were induced after 20 h of acclimation to VL-CO$_2$ in a CIA5-dependent manner, only photorespiratory proteins were detected under H-CO$_2$ (Fig. 3b, Supplementary Fig. 8), indicating that photorespiratory enzymes and CCM proteins are regulated differently at the protein level. Interestingly, photorespiratory proteins GCSP and GYD1 levels were more abundant in wild-type CC125 and *cia5* under VL-CO$_2$ at 21% O$_2$ compared to VL-CO$_2$ 2% O$_2$ whereas CCM related proteins were unaffected by O$_2$ levels (Fig. 3b, d, Supplementary Fig. 8). HPR1 abundance was only affected by O$_2$ levels in the *cia5* (Supplementary Fig. 8b). This effect of O$_2$ levels on the abundance of photorespiratory proteins was not observed in the wild-type CC4533 strain (Supplementary Fig. 7b), which could indicate a feedback regulation of photorespiratory proteins.

To better understand the reasons for the growth defect observed in *lci20*, and further determine whether it may be caused by the observed decrease of GYD1 abundance in *lci20* mutant (Fig. 3c), we have isolated

and characterized two CLiP mutants of *Chlamydomonas* harboring insertions in introns of the *GYD1* gene (Fig. 4a) and validated to be deficient in the GYD1 protein by immunoblot (Fig. 4b). Both mutants showed higher glycolate excretion than the control strain (Fig. 4c). However, the growth of both *gyd1* mutants was similar to the control wild-type strain after a transition from H-CO$_2$ to VL-CO$_2$ regardless of O$_2$ levels (Fig. 4d, Supplementary Fig. 9). *gyd1* mutants did not require H-CO$_2$ to grow, which is in contrast to a previous work where the H-CO$_2$ requiring 89 mutant (HCR89) is reported to harbor a mutation at the *GYD1* locus[45]. Therefore, the growth defect of *lci20* observed during H-CO$_2$ to VL-CO$_2$ is unlikely due to a decrease in GYD1 abundance but rather results from the accumulation of photorespiratory metabolites downstream the GYD1 reaction step. Indeed, the *lci20* mutants excreted half as much glycolate as *gyd1* mutants (Figs. 2b and 4c), which likely explain why their growth is more affected during a transition from H-CO$_2$ to VL-CO$_2$ under photorespiratory conditions.

## Photosynthesis of *lci20* is affected under photorespiratory conditions

To investigate the consequence of impaired photorespiration on the photosynthetic capacity of *lci20*, we performed room temperature chlorophyll fluorescence measurements to assess the PSII quantum yield and PQ redox state (1-qL) in H-CO$_2$ grown cells and following their acclimation to VL-CO$_2$ (Fig. 5, Supplementary Fig. 10). Under H-CO$_2$, *lci20* showed a slightly higher effective PSII quantum yield and lower PQ redox state compared to wild-type but was mainly similar to the complemented line (Fig. 5a, d). Following VL-CO$_2$ acclimation under photorespiratory conditions (21% O$_2$), *lci20* showed a significantly reduced effective PSII quantum yield and elevated PQ redox state compared to the control strains (wild-type and complemented line)

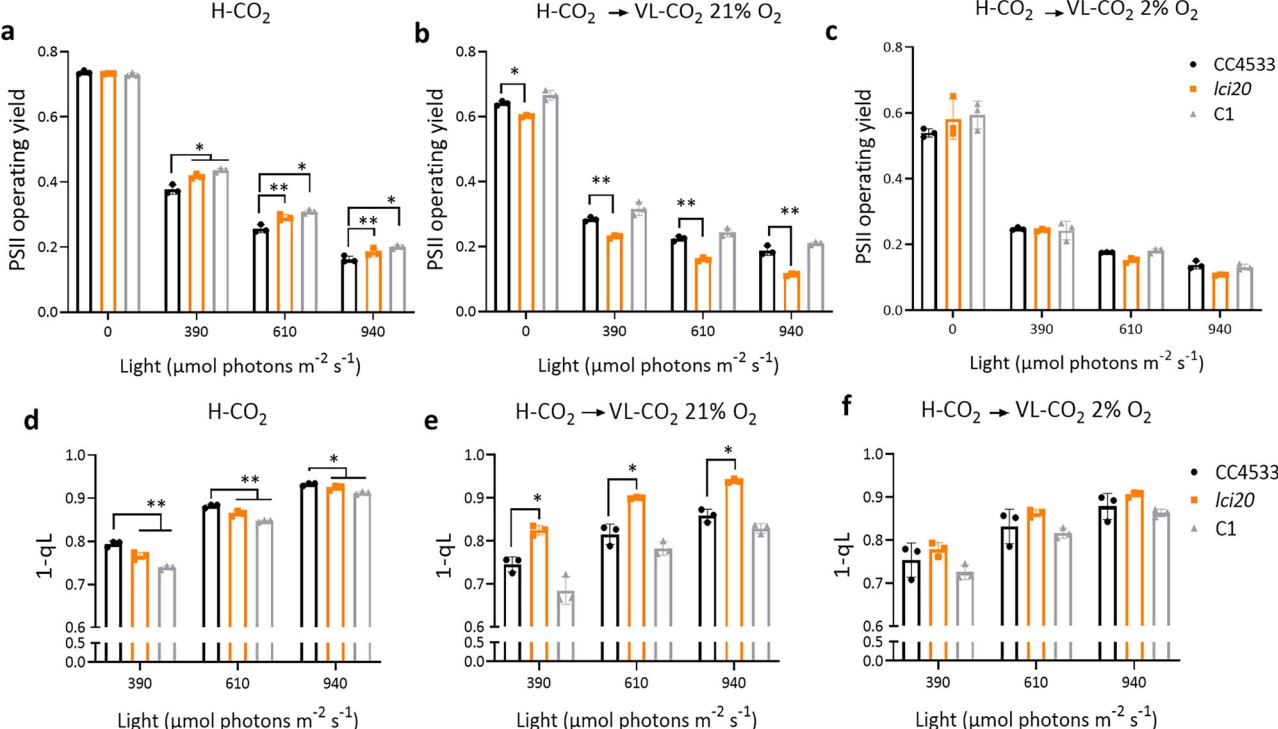

**Fig. 5 | Photosynthetic activity measured by chlorophyll fluorescence is reduced in *lci20* during acclimation to VL-$CO_2$ under photorespiratory (21% $O_2$) but not under non-photorespiratory (2% $O_2$) conditions.** PSII operating yields measured under actinic light in cells grown photo-autotrophically (**a**) under H-$CO_2$, (**b**) during acclimation from H-$CO_2$ to VL-$CO_2$ at 21% $O_2$ or (**c**) at 2% $O_2$ for 20 h. **d**–**f** PQ redox state measured as the (1-qL) parameter in the same samples as in (**a**–**c**). Cells were exposed for 1 min at the indicated light intensity. Bars show the average and dots show data from independent biological replicates ($n = 3 \pm SD$). Asterisks represent statistically significant difference compared to the wild-type strain CC4533 (*$p \le 0.05$ and **$p \le 0.01$) using two-way ANOVA.

(Fig. 5b, e). These effects were suppressed under non-photorespiratory conditions (VL-$CO_2$, 2% $O_2$) (Fig. 5c, f), thus indicating that the accumulation of non-metabolized photorespiratory compounds may be responsible for the inhibition.

## Metabolomic analyses reveal an increased accumulation of malate and glutamate in *lci20*

To investigate the metabolic changes occurring in *lci20* and obtain clues to the metabolites transported by LCI20, we performed a metabolomic analysis in the different strains grown under photoautotrophic H-$CO_2$, and during transitions from H-$CO_2$ to L-$CO_2$ and from H-$CO_2$ to VL-$CO_2$ (Fig. 6; Supplementary Fig. 11; Supplementary Data 1). The metabolite profile of H-$CO_2$-grown *lci20* cells was comparable to that of control strains (Fig. 6a Supplementary Fig. 11; Supplementary Data 1). After transition from H-$CO_2$ to L-$CO_2$ or to VL-$CO_2$, we observed a higher accumulation of malic acid and glutamic acid in *lci20* when cells were cultivated under photorespiratory condition (21% $O_2$), the effect being suppressed under non-photorespiratory conditions (2% $O_2$) (Fig. 6b, d). Accumulation of almost all the other metabolites detected was not significantly affected in *lci20* under the conditions tested (Supplementary Fig. 11; Supplementary Data 1). Based on these results, we propose that LCI20 is involved in the transport of malate and glutamate during photorespiration and that disruption of glutamate transport impairs photorespiratory glycolate metabolism at the step of glyoxylate conversion to glycine thus explaining the *lci20* phenotypes observed during acclimation to VL-$CO_2$.

## Discussion

In this study, we demonstrate the role of LCI20 in the photorespiratory glycolate metabolism in *Chlamydomonas* (Fig. 7), as evidenced by a high glycolate excretion and the marked decline in growth of the *lci20* mutant during a transition from H-$CO_2$ to VL-$CO_2$ at ambient $O_2$ levels

(21%), these effects being suppressed at 2% $O_2$ (Figs. 1, 2). Many photorespiratory mutants have been previously reported in land plants and only very few in algae. In *Arabidopsis*, a defect in genes encoding photorespiratory enzymes generate a high-$CO_2$ requiring phenotype, which is attributed to the accumulation of non-metabolized photorespiratory intermediates[13]. In *Chlamydomonas*, the situation is not that clear, and this is further complicated by the occurrence of CCM in algae. A phosphoglycolate phosphatase 1 (*pgp1*) mutant, impaired in the conversion of 2-PG to glycolate, exhibits a growth defect under stationary conditions of limiting $CO_2$ levels[24]. The high-$CO_2$ growth requirement of *pgp1* was attributed to the inhibitory effect of non-metabolized 2-PG accumulating upon ribulose bisphosphate oxygenation[24]. After the conversion of 2-PG into glycolate by PGP1, glycolate can follow two different fates in *Chlamydomonas*, it can be either excreted out of the cell or metabolized into glyoxylate by GYD1 in the mitochondria. A *Chlamydomonas gyd1* mutant isolated from a screening for an H-$CO_2$ requirement showed a high glycolate excretion during transition from H-$CO_2$ to L-$CO_2$[45]. However, the H-$CO_2$ requiring phenotype of this mutant was not clearly evidenced. We have shown here that impairing photorespiration at the level of GYD1 does not lead to an H-$CO_2$ requirement but instead to a high glycolate excretion under VL-$CO_2$ (Fig. 4, Supplementary Fig. 9). It was recently reported that growth of *Chlamydomonas hpr1* mutant is slightly impaired during the transition from H-$CO_2$ to L-$CO_2$, accompanied by a high glycolate excretion[57]. The slight growth defect of the *hpr1* mutant being interpreted as a consequence of glycolate hyper-excretion[57]. However, the *gyd1* mutants characterized here show high glycolate excretion rates but no growth defect in any $CO_2$ conditions tested (Fig. 4, Supplementary Fig. 9). We therefore conclude that inhibition of photorespiration at different steps of the pathway can lead to contrasting phenotypes. While inhibition of glycolate conversion does not impair growth (*i.e.* in *gyd1*), the accumulation of intermediate metabolites

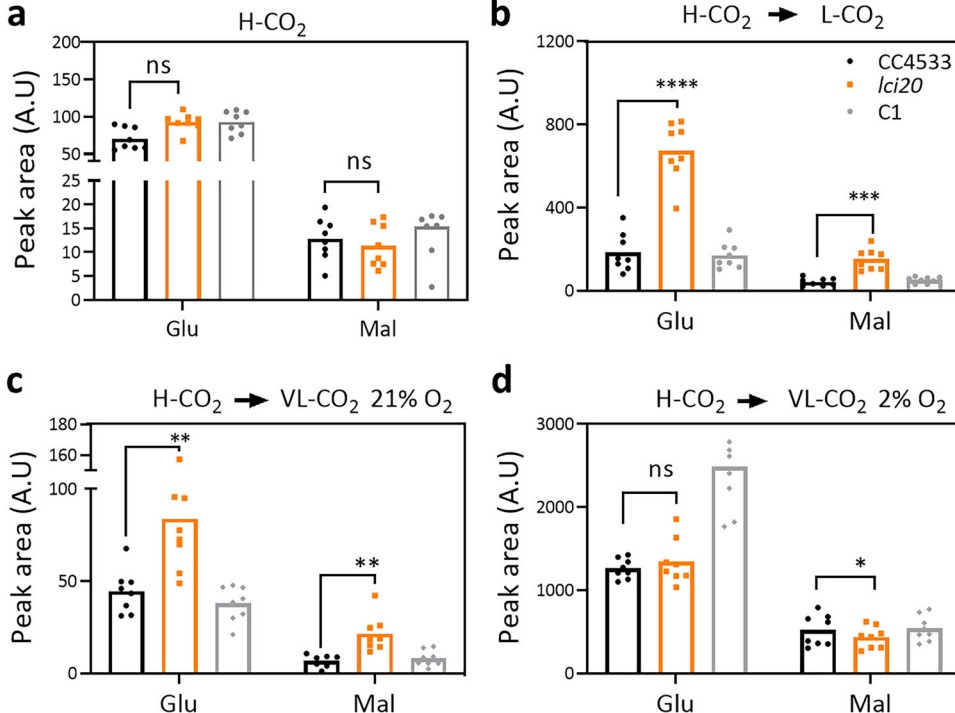

**Fig. 6 | Metabolomic analysis shows increased accumulation of malate and glutamate in *lci20* during a transition from H-CO₂ to VL-CO₂ under photorespiratory but not under non-photorespiratory conditions.** Accumulation of intracellular glutamic acid and malic acid in cells grown (**a**) under H-CO₂, (**b**) or after transition from H-CO₂ to L-CO₂ for 20 h, (**c**) after transition from H-CO₂ to VL-CO₂ under 21% $O_2$ or (**d**) under 2% $O_2$ for 20 h. Bars show the average and dots show data from independent biological replicates ($n = 8 \pm$ SD). Asterisks represent statistically significant difference compared to the wild-type strain CC4533 (*$p \leq 0.05$, **$p \leq 0.01$, ***$p \leq 0.001$, and ****$p \leq 0.0001$) using two-way ANOVA. A.U., arbitrary unit; ns, not significant.

such as 2-PG (*i.e.* in *pgp1*) or downstream glycolate conversion (i.e. in *hpr1*) is deleterious. The strong down-regulation of GYD1 observed in *lci20* under photorespiratory conditions (Fig. 3a) may result from a protective mechanism aiming at limiting the toxicity of non-metabolized photorespiratory intermediates by decreasing their production while promoting glycolate excretion. Consequently, glycolate excretion together with GYD1 down-regulation could be seen as complementary protection mechanisms preventing the toxic accumulation of photorespiratory intermediates during acclimation to VL-CO₂. Such mechanisms would be particularly needed until CCM components and photorespiratory enzymes are fully induced, thus allowing photorespiratory metabolism to be tuned to the residual Rubisco oxygenase activity which remains at VL-CO₂ despite the presence of CCM.

Unlike the above-mentioned proteins (PGP1, GYD1 and HPR1), which are all involved in the core photorespiratory pathways, LCI20 turns out to be an auxiliary protein involved in the supply of amino (-NH₂) donor as glutamate for the conversion of glyoxylate into glycine (Fig. 7). In *Arabidopsis* chloroplasts, the export of glutamate in counter-exchange with malate is mediated by the dicarboxylate transporter DiT2.1 that is required for photorespiratory nitrogen assimilation and displays a 60% homology with the *Chlamydomonas* LCI20[58–61]. Based on our metabolomic analysis, we suggest that LCI20 mediates the export of glutamate in counter exchange with malate across the chloroplast inner envelope (Fig. 6; Supplementary Fig. 11). Impairing glutamate export by knocking out LCI20 likely disrupts glycolate metabolism at the step of glyoxylate transamination. The higher glycolate excretion observed in *lci20* (Fig. 2b) would then result from an inefficient conversion of glycolate into glyoxylate, which would in turn result in a decrease in the GYD1 abundance (Fig. 3a, c). Both glycolate and glyoxylate, which could not be determined in our metabolomic analysis, are intermediate metabolites whose accumulation has been shown to be toxic for plants[62,63]. Because *Chlamydomonas* is unlikely to accumulate glycolate due to the existence of an excretion pathway, we

conclude that the growth impairment of *lci20* does not result from glycolate excretion per se, but rather from poisoning by non-metabolized glyoxylate. In line with our data, mutants deficient in the LCI20 homologues DiT2.1 and OMT1 in tobacco and Arabidopsis showed photoinhibition of PSII caused by the negative feedback of non-metabolized glyoxylate on the activation state of the Calvin cycle enzymes[60,64,65]. Glyoxylate was shown to be over accumulated in *omt1* mutants in both Arabidopsis and tobacco[60,64]. The negative effects of glyoxylate on the activation state of Rubisco was shown in several studies[66,67]. Overall, LCI20 plays a vital role during the acclimation of *Chlamydomonas* from H-CO₂ to VL-CO₂ by exporting glutamate from chloroplast toward mitochondria which is required for the function of photorespiratory pathway to efficiently recycle Rubisco oxygenation product 2-PG back to the Calvin cycle in the chloroplast.

The fact that *lci20* growth is unaffected under steady-state L-CO₂ or VL-CO₂ could be explained in two different ways. On the one hand, we could consider that photorespiration is only transiently active before the activation of CCM during acclimation to sub-optimal CO₂ levels but is suppressed or strongly reduced upon CCM activation under steady-state conditions. A similar conclusion was drawn from the observation that the glycolate excretion is suppressed in air acclimated *Chlamydomonas* wild-type cells when CCM is fully induced[18]. However, we can also imagine that the arrest of glycolate excretion is caused by the upregulation of enzymes of the glycolate metabolic pathway both at transcript and protein levels. The later hypothesis is supported by the fact that the transaminase inhibitor aminooxyacetate (AOA), which blocks the photorespiratory pathway, induces a higher glycolate excretion in air-adapted *Chlamydomonas* wild-type cells with a fully activated CCM[45]. Alternatively, we could consider that during photorespiration the need for LCI20-mediated -NH₂ recycling is only transient. Indeed, two -NH₂ are needed for the conversion of two molecules of glyoxylate into two glycines. One -NH₂ is likely supplied by the conversion of one serine into one

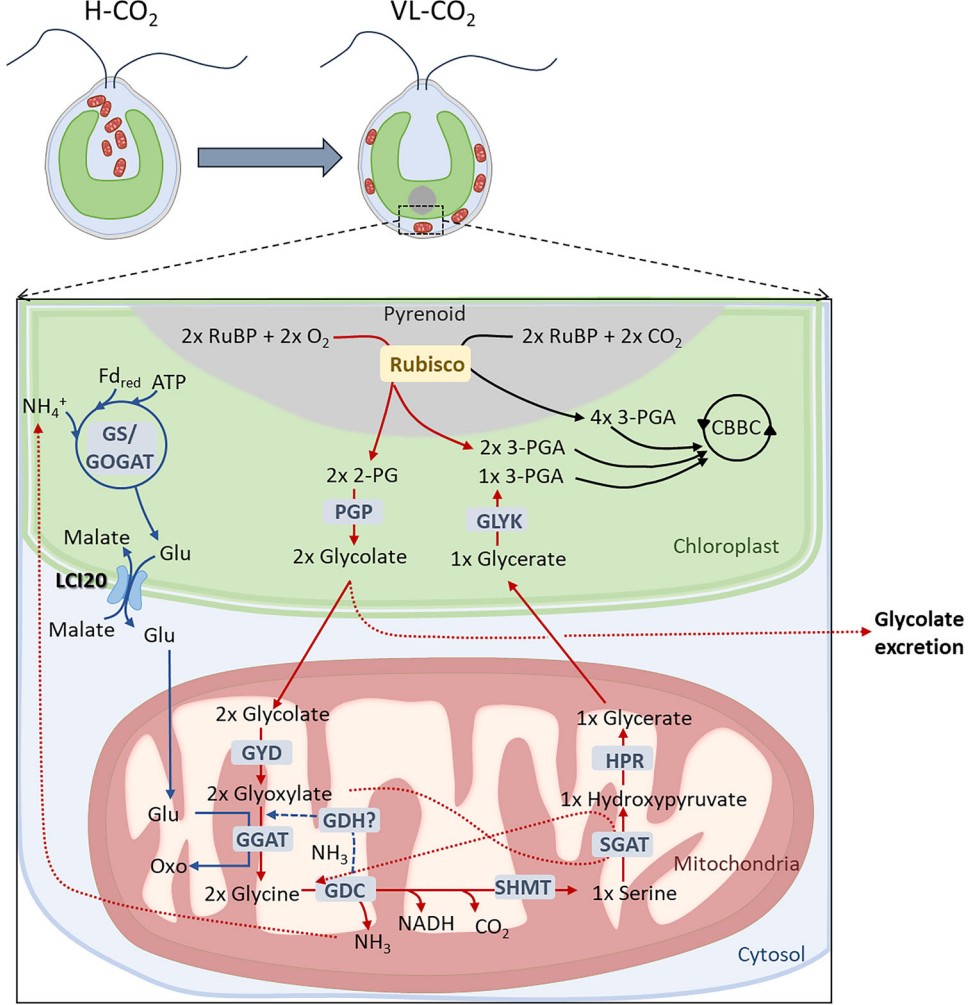

**Fig. 7 | Hypothetical scheme describing the role of LCI20 during photo-respiration in *Chlamydomonas*.** Photorespiration is initiated by the oxygenase activity of Rubisco producing phosphoglycolate (2-PG), which is converted into glycolate by a phospho-glycolate phosphatase (PGP), and then glyoxylate by the glycolate dehydrogenase (GYD). Glyoxylate is transaminated into glycine by a glutamate-glyoxylate aminotransferase (GGAT) using glutamate as a -NH₂ donor. During the condensation of two molecules of glycine into one serine by the glycine decarboxylase (GDC), CO₂ and NH₃ are released. Photorespiratory NH₃ is likely reassimilated by the GS/COGAT as proposed in *Arabidopsis*[59,87] and

*Chlamydomonas*[17,68]. LCI20 would function as a glutamate transporter exporting glutamate out of the chloroplast in counter-exchange with malate, thus supplying a -NH₂ donor for glyoxylate conversion into glycine. The growth phenotype of *lci20* observed during a transition from H-CO₂ to VL-CO₂ would result from the accumulation of glyoxylate. The absence of *lci20* phenotype under steady state L-CO₂ or VL-CO₂ would result from the induction of an alternative recycling pathway for the photorespiratory NH₃, for instance the mitochondrial glutamate dehydrogenase (GDH).

hydroxypyruvate. Since the conversion of two molecules of glycine into one serine is accompanied by the release of NH₃, the later needs to be re-assimilated to balance the nitrogen budget. In *Chlamydomonas*, inhibition of glutamine synthetase induces excretion of NH₃ in an O₂-dependent manner therefore showing that at least part of the NH₃ produced by photorespiration is recycled within chloroplasts by the GS-GOGAT pathway[68]. In *Arabidopsis*, the re-assimilation of photo-respiratory NH₃ is mediated by the chloroplast GS/GOGAT cycle, which produces glutamate in turn re-imported into mitochondria; this pro-cess is essential for photorespiration as evidenced by the lethal phe-notype of both chloroplast and mitochondrial glutamate transporters[59,69,70]. In this context, the transient phenotype of *lci20* may result from the induction of an alternative pathway of NH₃ reas-similation for instance through a mitochondrial isoform of the gluta-mate dehydrogenase[71] which may compensate for the lack of LCI20 under steady state limiting CO₂ (Fig. 7).

An hypothesis often put forward regarding the physiological role of photorespiration in microalgae is that photorespiratory metabolites could be the metabolic signal responsible for the induction of CCM

during acclimation to sub-optimal CO₂ levels[11,14,22–24,72,73]. Our data clearly establish that both the accumulation of CCM components (Fig. 3b, Supplementary Fig. 8) and the CCM functioning (Fig. 2a) are indepen-dent of O₂ levels, thus ruling out this hypothesis. In C3 plants, it has recently been proposed that photorespiration helps to maintain cellular redox balance by generating malate in the chloroplast, with malate export to the cytosol providing NADH for nitrogen assimilation[74]. Such a role in redox balance or any other physiological function of photo-respiration remains to be demonstrated in microalgae.

This work paves the way for a more detailed understanding of photorespiration in microalgae and how it interacts with the CCM functioning. Significant amounts of glycolate have been found in the ocean, particularly during algal blooms[75–79], suggesting that photo-respiration can be active in phytoplankton despite the occurrence of CCMs in most species[9,80,81]. This apparent discrepancy could be due to the fact that CCMs from different marine species show quite different efficiencies[9,80,81]. For instance, *Nannochloropsis* has been reported to harbour a quite leaky CCM[82]. It is also possible that algal species with an efficient CCM excrete significant glycolate amounts in particular

conditions, for instance when the $CO_2$ concentration drops during algal blooms, or when the incident light suddenly increases. Further studies are needed to better understand how algal photorespiration contributes to the carbon footprint of oceans.

## Methods

### Growth conditions and strains

The *lci20* (LMJ.RY0402.205944), *gyd1-1* (LMJ.RY0402.206272), *gyd1-2* (LMJ.RY0402.232044), *cia5* (CC-2702), and wild-types CC4533 and CC125 were purchased from the *Chlamydomonas* resource center. Cells were cultivated in an incubation shaker (INFORS Multitron pro) maintained at 25 °C, with 120 rpm shaking and constant illumination at 80 $\mu$mol m$^{-2}$ s$^{-1}$ supplied by fluorescent tubes delivering white light enriched in red wavelength. Cells were grown in MOPS-buffered (20 mM MOPS, pH 7.2) minimal medium (MM) exposed to various $CO_2$ levels. Photorespiration was induced by transferring 2% (H) $CO_2$-grown cells into 0.01% (VL) $CO_2$ under atmospheric $O_2$ level (21%) and can be suppressed by 2% $O_2$. Growth kinetics were monitored with a Multi-sizer 3 Coulter counter (Beckman Coulter). For the wild-type and *gyd1-1* growth assays performed in liquid culture, cells were cultivated photoautotrophically in 1-L photobioreactors operated as turbido-stats. Cell density was monitored through turbidity measurement (OD 800 nm) and maintained at 0.4 O.D by addition of fresh media. The temperature was maintained at 25 °C, the pH at 7.2, at the light intensity at 125 $\mu$mol photons m$^{-2}$ s$^{-1}$. Cultures were bubbled with gas mixtures containing $CO_2$ at 4%, 0.1%, or 0.04% in air, and supplied at a constant flow rate of 0.5 L min$^{-1}$. The dissolved $CO_2$ concentration in the photobioreactor medium was measured by membrane inlet mass spectrometry[83] upon rapid filtration of the culture. For spot test, cells were grown in liquid culture with MM, harvested during active growth at around 10 $\mu$g mL$^{-1}$ of chlorophyll and resuspended in fresh MM medium to make series of dilutions to 0.5, 1, and 2 $\mu$g mL$^{-1}$ chlorophyll per spot. Eight-microliter drops were spotted on 1.5% of MM agar plates at pH 7.2 buffered with 20 mM MOPS and exposed to various $CO_2$ and light regimes. Homogeneous light was supplied by a panel of fluorescent tubes.

### Protein extraction and immunoblot analysis

Total protein was extracted as previously described in ref. [26]. Exponentially grown cells (equivalent of 20 $\mu$g of chlorophyll) were harvested by centrifugation at 4000 $g$ for 3 min at 4 °C. Pellets were resuspended in 200 $\mu$L of PBS with a complete protease inhibitor EDTA-free mixture tablet (Roche) by vortexing and sonicated for 15 s 30% pulses on ice using a sonicator (product number UR-21P; TOMY). 200 $\mu$L of Novex™ Nupage™ LDS buffer 2x (Invitrogen™) containing 1x reducing agent DTT was added to the solution, and the total protein was solubilized by incubation at 37 °C for 20 min. Incubated samples were subsequently centrifuged at 4000 $g$ for 3 min. 10 $\mu$L (1X) of protein samples were loaded on Novex™ Nupage™ Bis tris 10% or tris-acetate 3-8% (Invitrogen™) gel, migrated 1 h at 190 V in Novex™ Nupage™ MOPS or tris-acetate (Invitrogen™) buffer according to protein molecular weight and transferred to nitrocellulose membrane using semidry transfer technique. Immuno-detection was performed using antibodies raised against LCI20 (1/250), GYD1 (1/500), HPR1 (1/500), GCSP (1/500), HLA3 (1/500), LCI1 (1/500) and LCIC (1/500). Antibody raised against α-Tubulin 1/2000 (Sigma-Aldrich ref. T6074) was used as control. Secondary anti-rabbit peroxidase-conjugated antibodies (Sigma-Aldrich; no. AQ132P) (1/10,000) were used for the detection with the G:BOX Chemi XRQ system (Syngene) using ECL detection reagents (GE Healthcare). Images were captured with a CCD camera equipped with a GeneSys Image Acquisition Software (Syngene).

### Glycolate quantification

Excreted glycolic acid was analyzed from the supernatant after centrifugation of 1 mL culture at 4000 $g$ for 2 min at 4 °C. The supernatant was diluted 10x in acetonitrile and analyzed using a Vanquish UHPLC/Q Exactive Plus (ThermoScientific). Glycolic acid was separated using a HILIC stationary phase (SeQuant ZIC HILIC, 100 × 2.1 mm, 3.5 $\mu$m), heated to 35 °C. A binary solvent system was used, in which mobile phase A consisted of acetonitrile: water (95:5, v/v) with 5 mM ammonium acetate and mobile phase B consisted of water:acetonitrile (95:5, v/v) with 5 mM ammonium acetate. Separations were made over a 32 min period following the gradient: 0-0.5 min: 5% B, 0.5–24.5 min: 5-95% B, 24.5–26.5 min: 95% B, 26.5–26.6 min: 95-5% B, 26.6–32 min: 5% B. The flow rate was set to 0.3 mL min$^{-1}$ and the injection volume was 5 $\mu$L. After separation, glycolic acid was directed into the ESI source of the Q Exactive Plus (Orbitrap-mass spectrometer). The ESI source was set as following: negative mode ion spray voltage at −2.5 kV, capillary temperature at 300 °C, S-lens RF level at 50. Data was acquired using a targeted t-SIM method with glycolic acid in inclusion list (formula $C_2H_4O_3$, m/z 75.00877).

### Measurement of chlorophyll fluorescence

Chlorophyll fluorescence measurements were performed using a Dual Pulse Amplitude Modulated Fluorometer (DUAL-PAM-100; Walz) equipped with a red LED source of actinic light. Samples (2 mL of cells at 10 $\mu$g mL$^{-1}$ chlorophyll) were placed into a cuvette under constant stirring at room temperature (25 °C) and dark adapted for 15 min prior to measurement. Both effective PSII quantum yield and 1-*qL* were calculated from the light curve. The latter was obtained by increasing red actinic light stepwise every 1 min starting from 17, 110, 190, 390, 610 and 940 $\mu$mol photons m$^{-2}$ s$^{-1}$, each being separated by a saturating pulse 10,000 $\mu$mol photons m$^{-2}$ s$^{-1}$, 600 ms duration. The effective PSII quantum yield and 1-qL were calculated as $\Phi PSII = (F_m\text{-}F_s)/F_{m'}$ and $qL = ((F_m\text{-}F_s)/(F_m\text{-}F_0))*(F_0/F_s)$ respectively with $F$m' the fluorescence value after saturating pulses, $F$s the stationary fluorescence at each actinic light and $F_0$ the fluorescence value in the dark[84].

### Metabolomics analysis

Cells were grown to exponential phase under photoautotrophic H-$CO_2$ conditions in the flask with 80 $\mu$mol photons m$^{-2}$ s$^{-1}$. For L-$CO_2$ and VL-$CO_2$ conditions, H-$CO_2$ grown cells were transferred to either L-$CO_2$ or VL-$CO_2$ levels for 20 h prior to sampling. About 60 million cell suspensions were injected into a −70 °C cold quenching solution composed of 70% methanol in water using a thermoblock above dry ice to obtain a final concentration of 35% methanol. Centrifuge tubes containing the cell suspensions were cooled in a pre-chilled cooling box to keep the sample temperature below −20 °C. Cell pellets were collected by centrifugation at 4000 $g$ for 2 min at −10 °C. The supernatant was decanted and residual liquid together with cells carefully transferred to a 2 mL Eppendorf tube and re-centrifuged 13,000 $g$ for 1 min at −10 °C. The pellet was flash-frozen in liquid nitrogen and lyophilized at −50 °C. The extraction and derivation of the metabolites were performed as previously[85]. The analytic and quantification methods were exactly as reported in ref. [37]. Data is reported following recently updated metabolomics standards[86].

### Statistics

All statistical tests used are noted in figure legends. One-way or two-way ANOVA using GraphPad Prism (GraphPad Software) was used to perform statistical analysis. The *P*-values were computed by One-way or two-way ANOVA test (uncorrected $p$ values). Statistical significance ($\alpha = 0.05$) according to $p$ values is indicated by asterisk (*$p \le 0.05$; **$p \le 0.01$; ***$p \le 0.001$ and ****$p \le 0.0001$).

### Reporting summary

Further information on research design is available in the Nature Portfolio Reporting Summary linked to this article.

## Data availability

Genes studied in this article can be found on Phytozome 13 (Chlamydomonas genome v6.1 or 5.6) under the loci Cre06.g260450 [LCI20], Cre06.g295450 [HPR1], Cre06.g288700 [GYD1], Cre12.g534800 [GCSP], Cre03.g162800 [LCI1], Cre02.g097800 [HLA3], Cre06.g307500 [LCIC] and Cre02.g096300 [CIA5]. All antibodies used are available upon request to corresponding authors. Metabolites data are provided as supplementary data 1. All the other methods are reported in the Supplementary Information file . Source data are provided with this paper.

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

## Acknowledgements
O.D. thanks the French Atomic Energy and Alternative Energy Commission (CEA) for a PhD scholarship. We thank Shiyan Zheng for assistance in performing the genetic complementation experiment; Gaurav Kumar for technical assistance during microscopic observation; Stephanie Blangy for trials on generating antibodies against LCI20; Franziska Kuhnert for trials on the transport assays. We thank the ZoOM Microscopy facility (CEA Cadarache) and the University of York Biosciences Technology Facility for confocal microscopy access. This work is financially supported by the "L'économie Circular du Carbone" program of the CEA (CO2storage), the ANR project "AlgalCCM" (n°ANR-366 22-CE44-0023-01), the France 2030 initiative project "CO2_CMPhi" (ANR-23-PEXF-0002) and the Region Sud ("AlgalCO2" project). We acknowledge the European Union Regional Developing Fund (ERDF), the Région Provence Alpes Côte d'Azur, the French Ministry of Research and the CEA for funding the HelioBiotec platform. C.M. is grateful to the INRAE MIGALE bioinformatics facility (MIGALE, INRAE, 2020) for providing computing resources. AKL was supported by the Labex Saclay Plant Sciences-SPS (ANR-17-EUR-0007), the platform of Biophysics of the I2BC supported by the French Infrastructure for Integrated Structural Biology (FRISBI; grant number ANR-10-INSB-05). A.P.M.W. acknowledges funding by the European Union's H2020 research and innovation and the Horizon programs (grants GAIN4CROPS, GA No. 862087 and BEST-CROP, GA No. 101082091) and the Deutsche Forschungsgemeinschaft (Cluster of Excellence for Plant Sciences (CEPLAS) under Germany's Excellence Strategy EXC-2048/1 under project ID 390686111). S.A. and A.R.F. acknowledge the European Union's Horizon 2020 research and innovation program, project PlantaSYST (SGA-CSA No. 739582 under FPA No. 664620) and the BG05M2OP001-1.003-001-C01 project, financed by the European Regional Development Fund through the Bulgarian 'Science and Education for Smart Growth' Operational Program. Both authors acknowledge the support by the Max Planck Society. A.B. acknowledges support from the Carnegie Institution for Science.

## Author contributions
Y.L.-B., G.P., O.D. conceived the study. Y.L-B. and G.P. supervised the work. O.D. performed most of the experiments. A.B., A.K.L. and G.P. supervised photosynthesis measurement performed by O.D. P.A., M.B., S.C. and O.D. carried out immunoblots. O.D. and V.E. isolated and characterized *gyd1* mutants with contribution from A.M. C.M. performed phylogenetic analysis. M.B., F.V. and O.D. performed genetic complementation of *lci20* mutant. S.A. and A.R.F. performed metabolomics. O.D., P.-C.N. and B.L. performed glycolate analysis. O.D., G.P. and Y.L.-B. drafted the manuscript with contributions from L.C.M.M., A.B., A.K.L., A.R.F. and A.P.W.

## Competing interests
The authors declare no competing interests.
