## [Transparent Peer Review file · Nature Communications]

The green algae CO₂ concentrating mechanism and photorespiration jointly operate during acclimation to low CO₂

Corresponding Author: Dr Yonghua Li-Beisson

Version 0:

Reviewer comments:

Reviewer #1

(Remarks to the Author)

This study presents convincing evidence that in the microalga *Chlamydomonas reinhardtii* (1) CCM induction does not depend on photorespiration and (2) photorespiration is active at low CO₂ when the CCM is operational. This evidence is based on the influence of a mutation in a chloroplast-envelope glutamate/malate transporter (low-CO₂ inducible 20, LC120), under different CO₂ and O₂ treatments that stimulate or inhibit carbon fixation or photorespiration. These well-planned experiments were conducted using standard procedures. The authors, however, might consider some additions to the manuscript that would enhance the accessibility of this research to the broader audience of Nature Communications.

In particular, the introduction might include information comparing the biochemistry of photorespiration in *Chlamydomonas* with that in vascular plants because *Chlamydomonas* does not have a peroxisome. In vascular plants, oxygenation of RuBP generates 2PG plus 3PGA, and then convert 2PG into glycerate through a series of reactions in peroxisomes and mitochondria; in *Chlamydomonas*, analogous reactions occur solely in mitochondria. Perhaps as a result, *Chlamydomonas* is normally dependent on a CCM, has distinct responses to H-CO₂, L-CO₂, and VL-CO₂ treatments, and sometimes excretes glycolate,

On p. 14, the manuscript states that both glycolate and glyoxylate could not be determined in the metabolomic analysis. This is because the metabolomics in this part of the study depended on analyses based on GC-MS which suffers from peak-tailing (Arrivault et al. 2009. *The Plant Journal* 59:826-839) and matrix effects (Tarakhovskaya et al. 2023. *Molecules* 28(6):2653, doi: 10.3390/molecules28062653; Alseekh et al. 2021. *Nature Methods* 18:747-756). Metabolomics based on LC-MS methods avoid such difficulties Koley et al. 2021. *Journal of Experimental Botany*, 73(9), 2938-2952. doi:10.1093/jxb/erac062). These were used in a different portion of this study for glycolate quantification (p. 18).

The data in this study do not support the hypothesis that photorespiratory metabolites are responsible for the induction of CCM during acclimation to sub-optimal CO₂ levels (p. 16). What do the authors propose as the physiological role of photorespiration in microalgae? Might photorespiration generate malate in chloroplasts? We recently proposed a photorespiration pathway in C3 plants whereby (1) rubisco when associated with manganese converts a substantial amount of RuBP into pyruvate, (2) malic enzyme when associated with manganese carboxylates a substantial portion of this pyruvate into malate, and (3) chloroplasts export additional malate into the cytoplasm where it generates NADH for assimilating nitrate and sulfate into amino acids (Shi, Hannon, & Bloom 2024. *Physiologia Plantarum* 176:e14463.doi:DOI: 10.1111/ppl.14463).

The scheme depicted in Figure 7 is confusing. The reaction for RuBP oxygenation is $2 \text{ RuBP} + \text{O}_2 \text{ yields } 2\text{PG} + 2 \times 3\text{PGA}$. The reaction for RuBP carboxylation is $2 \text{ RuBP} + \text{CO}_2 \text{ yields } 4 \times 3\text{PGA}$.

Reviewer #2

(Remarks to the Author)

In this manuscript, Dao et al. attempt to explore the relationship between CO₂ concentrating mechanism and photorespiration in green algae under limited CO₂ conditions. They characterized either CCM- or photorespiration-deficient mutants by monitoring their growth and glycolate excretion under various photorespiration regimes. They propose that LCI20 is modulated by light and CO₂ levels as a candidate chloroplast malate transporter. The growth and photorespiratory glycolate metabolism of lci20 are impaired during transition from high to very low CO₂, likely due to the accumulation of photorespiratory metabolites downstream the GYD1 reaction step. Based on the metabolomic analysis, they propose that LCI20 may participate in supplying amino groups for the mitochondrial conversion of glyoxylate into glycine during photorespiration. This work provides some understanding of photorespiration and how it interacts with the CCM functioning in microalgae.

However, the data presented here are not yet solid and convincing enough to support the conclusion. Considering that the underlying mechanism has not been revealed in the study, the paper could be submitted to a more specialized journal after substantial and careful revision. The comments are outlined below.

Major concerns:

1. Since we know that CCM could be induced by the low CO₂ condition and photorespiration is affected by the ratio of O₂/CO₂, it's not difficult to get the speculation in lines 44-48: "By monitoring growth.....when the CCM is operational". In fact, studies have already been reported that glycolate excretion protects algal cells from the toxicity of excess photorespiratory intermediates. Thus, the conclusion asserted here appears to be another piece of evidence for previous reports, which affects the novelty of the study.
2. To make the results of this study convincing, a control strain deficient in regular low carbon induced protein is required to distinguish the function of LCI20 from other LCI proteins. Especially in the experiment with both O₂ and CO₂, the control strain is necessary to clarify the function of LCI20.
3. If LCI20 is responsible for the transport of malic acid and glutamic acid across the chloroplast envelope, their accumulation in specific subcellular regions, e.g., chloroplast or cytoplasm, should be measured to evaluate the function of LCI20 in metabolomic analysis. The overall accumulation detected in this study may not reflect the true function of LCI20.
4. The quality of data presented here is not convincing enough to support the claims, e.g. western blot. Thus, further propose based on the western blot result remains to be confirmed.

Other comments

1. Line 128-130: Since CrLCI20 was annotated as the chloroplast oxoglutarate/malate transporter based on the bioinformatics ortholog analysis, experimental data are required to confirm its function by measuring and comparing the malate concentration in the chloroplast of WT and lci20 strains.
2. L140-L143; L150-151: Considering that LCI20 could be regulated by the upstream transcription factor, the authentic promoters rather than PSAD promoter should be used in all of the experiments to avoid the false-positive results.
3. L171-172: The study here mentioned the photorespiratory (21% O₂) or non-photorespiratory (2% O₂) conditions for the first time. The experimental conditions used should be validated. Since the ratio of (O₂/CO₂) is still very high under 2%O₂/VL-CO₂ condition, the non-photorespiratory conditions used here may not be convincing.
4. Line177-180 "Note that similar growth was observed when wild-type cells were exposed to VL-CO₂ either under photorespiratory conditions (21% O₂) or non-photorespiratory conditions (2% O₂), indicating that photorespiration is not required for CCM induction". To confirm this, liquid culture results are required and are more convincing than solid plate culture by measuring the growth performance of the strains.
5. Line190-192: "We conclude that lci20 is defective in the photorespiratory glycolate metabolism, which is triggered when Chlamydomonas cells are transferred from H-CO₂ to VL-CO₂." What about the phenotype of the regular lci mutant? Because of the high ratio (O₂/CO₂), glycolate excretion could also be induced in the regular lci mutant. Therefore, it's necessary to include the regular lci mutant as a control strain to demonstrate the function of LCI20 in
6. Line201-206: In general, the quality of the Western blot data is not convincing to support the hypothesis stated in the text. Upon careful examination, GCSP and HPR1 share the similar profiles to that of GYD1 under 21% O₂ or even under 2% O₂ condition when comparing the lci20 and WT strain. To validate the results, multiple replicates of Western blot should be performed, and the quantification of the abundance profile of the target protein could be done using Image J software.
7. Line221-223: "However, the growth of both gyd1 mutants was similar to the control wild-type strain after a transition from H-CO₂ to VL-CO₂ regardless of O₂ levels (Fig. 4d)". To validate the results, growth data from the liquid culture is required and the growth profile on the solid plate is not sufficient to draw a conclusion.
8. Line232-245: Section "Photosynthesis of lci20 is affected under photorespiratory conditions" could be moved to the supplemental materials, as several studies have already reported the impairment of photosynthesis by photorespiration defects.

Therefore, the reviewer would not recommend this manuscript to be considered or published in the Nature Communication. It could be submitted to a specialized journal after careful revision.

Reviewer #3

(Remarks to the Author)

This paper reports a large effort to better understand photorespiration in organisms that have CO₂ concentrating mechanisms, specifically Chlamydomonas. The conclusions are well supported and are a significant step forward. Their methods are described in good detail. All data are provided in supplementary files. I don't have any substantive concerns but can point out some trivial things for the authors' consideration.

1. It took me some time to work out the name of the gene and protein. Because of the similarity of capital I, I and lower case I, I was confused for a while I figured out that it was “low (L) inorganic carbon (Ci, here as Cl) 20, LCI20. But is it Ci or specifically CO₂? Help here would save readers time.
2. The paragraph on pages five and six is very long. Can it be broken up? Perhaps here “membrane²⁶. The contribution” And or transporters⁴³. By another way
3. Line 196, “Over excretion” sounds funny to me. How about “and large (or significant) glycolate excretion.”
4. Line 345 “decarboxylation of two molecules of glycine ...” gives the wrong impression. How about “conversion of two molecules of glycine to one serine...”
5. Line 347 this enzyme requires ATP and so it is a synthetase, synthases do not use ATP
6. Line 416 Glycolic acid was separated
7. 438 and elsewhere. I suggest italicizing all variables (as has been done for the different fluorescence parameters. That means qL should also be italicized. Also, so many of the qualifiers - m' and s of Fm' and Fs - should be subscripted. Perhaps eventually the field will eliminate these rules but for now I suggest sticking with them.
8. The references have a lot of issues with species names not italicized, capitalization of titles vary, subscripting is often absent. If the journal is going to reset the references from a database that will fix these then there is no concern but if the manuscript will be used to get these things right then some work needs to be done.
9. Supplementary Fig 1a. The expression at end of the dark period increases from 20% to 100 % between the end of the night on the right side to the beginning of day at the left side.

Version 1:

Reviewer comments:

Reviewer #1

(Remarks to the Author)

The authors have addressed my scientific concerns in the revised manuscript. In the attached annotated version of the manuscript, I have made several suggestions that might clarify the language in the article.

Reviewer #3

(Remarks to the Author)

The authors addressed all of my concerns.

Reviewer #1 (Remarks to the Author):

This study presents convincing evidence that in the microalga *Chlamydomonas reinhardtii* (1) CCM induction does not depend on photorespiration and (2) photorespiration is active at low CO₂ when the CCM is operational. This evidence is based on the influence of a mutation in a chloroplast-envelope glutamate/malate transporter (low-CO₂ inducible 20, LCI20), under different CO₂ and O₂ treatments that stimulate or inhibit carbon fixation or photorespiration. These well-planned experiments were conducted using standard procedures. The authors, however, might consider some additions to the manuscript that would enhance the accessibility of this research to the broader audience of Nature Communications.

In particular, the introduction might include information comparing the biochemistry of photorespiration in *Chlamydomonas* with that in vascular plants because *Chlamydomonas* does not have a peroxisome. In vascular plants, oxygenation of RuBP generates 2PG plus 3PGA, and then convert 2PG into glycerate through a series of reactions in peroxisomes and mitochondria; in *Chlamydomonas*, analogous reactions occur solely in mitochondria. Perhaps as a result, *Chlamydomonas* is normally dependent on a CCM, has distinct responses to H-CO₂, L-CO₂, and VL-CO₂ treatments, and sometimes excretes glycolate,

R: We would like to thank the reviewer for the positive comment and for bringing this issue to our attention and it is indeed an intriguing question. Actually, *Chlamydomonas* does have a kind of “peroxisome” where fatty acid beta-oxidation (Kong et al. 2018, *The Plant Cell*) and glyoxylate cycle have been shown to operate (Lauersen et al. 2016, *Algal Research*). Based on known literature, the function of the glycolate oxidase in *Chlamydomonas* remains elusive; it is generally considered that this reaction occurs in the mitochondria and is catalyzed by the glycolate dehydrogenase (Nakamura *et al.* 2005 and this study). In order to make this point clearer to a broad audience, we have now added a few sentences in the introduction: “Although photorespiration occurs in both plants and microalgae, their subcellular organization differs. In vascular plants, photorespiration requires tight cooperation of chloroplasts, peroxisomes and mitochondria, whereas in *Chlamydomonas*, photorespiration would mainly rely on chloroplast and mitochondria” (Eisenhut et al. 2019; Santhanagopalan et al. 2021; Shi et al. 2021, *Front Plant Sci.*)

On p. 14, the manuscript states that both glycolate and glyoxylate could not be determined in the metabolomic analysis. This is because the metabolomics in this part of the study depended on analyses based on GC-MS which suffers from peak-tailing (Arrivault et al. 2009. *The Plant Journal* 59:826-839) and matrix effects (Tarakhovskaya et al. 2023. *Molecules* 28(6):2653, doi: 10.3390/molecules28062653; Alseekh et al. 2021. *Nature Methods* 18:747-756). Metabolomics based on LC-MS methods avoid such difficulties Koley et al. 2021. *Journal of Experimental Botany*, 73(9), 2938-2952. doi:10.1093/jxb/erac062). These were used in a different portion of this study for glycolate quantification (p. 18).

R: We agree with the reviewer that metabolomics using GC-MS may have certain limitations as indicated above. However, the main problem why we were unable to measure glycolate and glyoxylate in our samples is due to the low quantity of these intermediate metabolites, which prevented their accurate quantification. We have lately attempted to solve this problem by increasing the amount of starting material but still failed to detect them.

Previous work aiming at quantifying glyoxylate and glycolate by LC-MS based targeted metabolomics have failed both in tobacco (doi.org/10.1093/jxb/erac062) and *Chlamydomonas* (DOI:10.1021/acs.est.0c08416). For example, Koley et al. (doi.org/10.1093/jxb/erac062) stated that “The peak intensities of photorespiratory intermediates such as 2-phosphoglycolate and glyoxylate were present at low levels on the hybrid column; however, the current approach can detect these peaks sensitively due to reduced background while the peaks were frequently indistinguishable from noise using the HILIC column.” In addition, it is generally accepted that the assessment of central

carbon metabolites in plants/algae can be challenging due to the dynamic range of pool sizes, making it difficult to quantify all metabolites from a single series using a single method.

Glycolate accumulation is unlikely to prevail in *Chlamydomonas* due to the existence of an excretion pathway. Regarding glyoxylate, its accumulation has been shown to be toxic for plants (Lu *et al.* 2014, *Plant Physiol.* 150, 463–476; Deller *et al.* 2016, *J. Exp. Bot.* 3041–3052), indicating that regulatory mechanisms likely limit its accumulation. These could be the reasons why we could not quantify these low abundant compounds using GC-MS.

Nevertheless, we are confident that the lack of data on glycolate and glyoxylate does not affect our conclusion. Indeed, in line with our conclusion, mutants deficient in the LCI20 homologues DiT2.1 and OMT1 in tobacco and *Arabidopsis* showed photoinhibition caused by the negative feedback of non-metabolized glyoxylate on the activation state of the Calvin cycle enzymes (Schneidereit *et al.*, 2006; Takahashi *et al.* 2007; Kinoshita *et al.* 2011). Glyoxylate was shown to be overaccumulated in *omt1* mutant in both *Arabidopsis* and tobacco (Schneidereit *et al.*, 2006; Kinoshita *et al.* 2011). The negative effects glyoxylate the activation state of Rubisco was shown in several studies (Campbell and Ogren, 1990; Häusler *et al.* 1996).

The data in this study do not support the hypothesis that photorespiratory metabolites are responsible for the induction of CCM during acclimation to sub-optimal CO₂ levels (p. 16). What do the authors propose as the physiological role of photorespiration in microalgae? Might photorespiration generate malate in chloroplasts? We recently proposed a photorespiration pathway in C3 plants whereby (1) rubisco when associated with manganese converts a substantial amount of RuBP into pyruvate, (2) malic enzyme when associated with manganese carboxylates a substantial portion of this pyruvate into malate, and (3) chloroplasts export additional malate into the cytoplasm where it generates NADH for assimilating nitrate and sulfate into amino acids (Shi, Hannon, & Bloom 2024. *Physiologia Plantarum* 176:e14463.doi:DOI: 10.1111/ppl.14463).

R: Thank you for mentioning this point. We have now discussed a possible role of algal photorespiration related to the production of malate in the chloroplast (L372-376) as follows: “In C3 plants, it has recently been proposed that photorespiration contributes to maintaining the cellular redox balance by generating malate in the chloroplast, with malate export to the cytosol providing NADH for nitrogen assimilation (Shi *et al.* 2024). Such a role in redox balance or any other physiological function of photorespiration remains to be demonstrated in microalgae”.

The scheme depicted in Figure 7 is confusing. The reaction for RuBP oxygenation is $2 \text{ RuBP} + \text{O}_2 \text{ yields } 2\text{PG} + 2\times 3\text{PGA}$. The reaction for RuBP carboxylation is $2 \text{ RuBP} + \text{CO}_2 \text{ yields } 4\times 3\text{PGA}$.

R: We have modified the scheme to introduce some stoichiometry for carboxylation and oxygenation (see Fig. 7).

Reviewer #2 (Remarks to the Author):

Major concerns:

1. Since we know that CCM could be induced by the low CO₂ condition and photorespiration is affected by the ratio of O₂/CO₂, it's not difficult to get the speculation in lines 44-48: "By monitoring growth.....when the CCM is operational". In fact, studies have already been reported that glycolate excretion protects algal cells from the toxicity of excess photorespiratory intermediates. Thus, the conclusion asserted here appears to be another piece of evidence for previous reports, which affects the novelty of the study.

R: We would like to point out here that not only one, but three different conclusions were drawn in lines 44-48. The first is that “CCM induction does not depend on photorespiration”, the third that “photorespiration is active at low CO₂ when the CCM is operational”, which both are clearly new findings. Concerning the second conclusion that “glycolate excretion protects algal cells from the toxicity of unmetabolized photorespiratory intermediates”, to our knowledge this has not been clearly

demonstrated in the literature. Unlike plants, very few algal photorespiratory mutants have been characterized so far. If the *Chlamydomonas* mutant deficient in phospho-glycolate phosphatase (Suzuki et al. 1990 doi: 10.1104/pp.93.1.231) shows a clear photorespiratory phenotype (need high CO₂ for growth), the growth inhibition is due to the toxicity of the unmetabolized phospho-glycolate accumulating in photorespiratory conditions, but this mutant neither produces nor excretes glycolate. A *Chlamydomonas* glycolate dehydrogenase (GYD1) mutant was isolated as a CO₂-requiring mutant (Nakamura et al. 2005 doi:10.1139/b05-067), indicating that the capacity to excrete glycolate does not protect from inhibition. In sharp contrast, we have shown here that two independent GYD1 mutants do not show any growth phenotype whatever the photorespiratory regime (**Fig. 4 and Supplementary Fig. 9**), which is clearly a novel finding and further indicates that the capacity to excrete non-metabolized glycolate prevents a photorespiration phenotype. Moreover, we have observed that photosynthesis of the *lci20* mutant is inhibited during low CO₂ acclimation (**Fig. 5**), similar phenotype being observed in recently characterized *Chlamydomonas* knock-down mutants of hydroxy-pyruvate reductases (Shi et al. 2021 doi: 10.3389/fpls.2021.690296), which showed some growth defect at low CO₂. Therefore, if glycolate excretion is sufficient to prevent inhibition of growth and photosynthesis in glycolate dehydrogenase-deficient mutants, the accumulation of other photorespiratory metabolites may generate some phenotype in mutants, like *lci20*, affected in downstream photorespiratory enzymes. Also, the strong down-regulation of GYD1 observed in *lci20* under photorespiratory conditions (Fig. 3a) may result from a protective mechanism aiming at limiting the toxicity of non-metabolized photorespiratory intermediates by decreasing their production while promoting glycolate excretion. In order to make the point clearer to the reader we have modified the second conclusion of the abstract as follows “glycolate excretion together with glycolate dehydrogenase down-regulation prevent the toxic accumulation of non-metabolized photorespiratory metabolites.” and further discuss the issue more in detail in the discussion (lines 46-48 and 327-337).

2. To make the results of this study convincing, a control strain deficient in regular low carbon induced protein is required to distinguish the function of LCI20 from other LCI proteins. Especially in the experiment with both O₂ and CO₂, the control strain is necessary to clarify the function of LCI20.

R: The term LCI refers to genes whose expression has been reported to be induced during low CO₂ acclimation (see Table 2 in <https://doi.org/10.1104/pp.107.114652>). They include genes involved in multiple mechanisms such as CCM, photorespiration, DNA binding, photoprotection, starch metabolism... In this study, we have used the *cia5* mutant as a control CCM deficient strain. CIA5 is known as the master regulator of most CCM related genes. To address this reviewer's concern, we have now performed additional experiments on two other mutants impaired in CCM components, the bestrophin-like bicarbonate channels BST1-3 (previously called LCI11C, LCI11A, LCI11B) and the thylakoid carbonic anhydrase CAH3 (now shown as **Supplementary Fig. 6**). In the same way as for *cia5*, both mutants did not grow during acclimation to VLC either at 21% O₂ or at 2% O₂, therefore consolidating our conclusion that LCI20 is not involved in CCM, but rather contributes to photorespiration (since *lci20* shows growth defect during acclimation to VLC at 21% O₂ but not at 2% O₂).

3. If LCI20 is responsible for the transport of malic acid and glutamic acid across the chloroplast envelope, their accumulation in specific subcellular regions, e.g., chloroplast or cytoplasm, should be measured to evaluate the function of LCI20 in metabolomic analysis. The overall accumulation detected in this study may not reflect the true function of LCI20.

R: We share the opinion of this reviewer on the significance of organelle specific based metabolomics, which gives more information compared to the whole-cell metabolomics approach. However, there are many challenges for organelle metabolomics including the limited availability of reliable methods for effective isolation of cellular compartments and reliable metabolite quantification within them (More & Hiller 2022 doi.org/10.1016/j.copbio.2022.102711). The determination of metabolites in purified chloroplasts makes it difficult to access the concentrations actually present before purification of the organelle. Actually, organelle isolation can be time consuming, during this process, metabolites

could have already been turned-over, and there is no guarantee that the level we measure in the isolated fraction reflects the amount present in the cell before organelle purification. For whole-cell metabolomics, there is a quick quenching step where cells were plunged into -70°C cold methanol (70%) to stop turnover. Particularly, in *Chlamydomonas*, it is very difficult to obtain intact chloroplasts although few studies have managed to do it in low amount (the average yield of intact chloroplasts being 13% as compared with whole cells based on chlorophyll recoveries) (Mason et al. 2006, doi:10.1038/nprot.2006.348). This low yield makes it difficult to effectively quantify metabolites using GC-MS. Furthermore, there are concerns with the chloroplast purity, as the isolated chloroplast fractions can still be contaminated with cytosolic or mitochondria components (Mason et al. 2006). Our whole-cell metabolomics showed overaccumulation of malate and glutamate in the *lci20* mutant which was fully recovered in the complemented lines and by growing under non-photorespiratory conditions (Fig. 6). Therefore, we are confident that such accumulations result from the defect of LCI20 and the impaired photorespiration in the *lci20* mutant.

4. The quality of data presented here is not convincing enough to support the claims, e.g. western blot. Thus, further propose based on the western blot result remains to be confirmed.

R: We have now performed additional western blot analysis from three independent biological replicates and further quantified the proteins abundance. These new data, supplied in **Supplementary Fig. 7 and 8** have been used to update Fig. 3. We have now performed statistical analysis for every immunoblots and further provided the original images as source data. The new data not only validate the previous ones but provide additional information. Interestingly, we have observed that some photorespiratory proteins (GCSP and GYD1 and to a lesser extent HPR1) are more abundant under photorespiratory condition than under non-photorespiratory condition. The effect of O₂ on photorespiratory proteins is only observed in the CC125 background (**Supplementary Fig. 8**).

Other comments

1. Line 128-130: Since CrLCI20 was annotated as the chloroplast oxoglutarate/malate transporter based on the bioinformatics ortholog analysis, experimental data are required to confirm its function by measuring and comparing the malate concentration in the chloroplast of WT and *lci20* strains.

R: Actually, the proposed function for CrLCI20 is not only based on bioinformatics ortholog analysis, but is also based on the subcellular localization (**Fig. 1a**). The clear chloroplast localization of LCI20 in *Chlamydomonas*, together with the high similarity with well-characterized plant glutamate/malate transporters such as DiT2, and the whole-cell metabolomics data showing overaccumulation of malate and glutamate in the *lci20* mutant are consistent with the function of LCI20 as a chloroplast glutamate/malate transporter. Of course, we would have liked to have specific metabolite data for each organelle, in particular for the chloroplast, but as mentioned in point 3 above, it is currently very challenging to measure these metabolites quantitatively, particularly in the case of *Chlamydomonas*.

2. L140-L143; L150-151: Considering that LCI20 could be regulated by the upstream transcription factor, the authentic promoters rather than PSAD promoter should be used in all of the experiments to avoid the false-positive results.

R: We agree that the native promoter would have probably been preferable, and there is no ideal solution here since even if we use the native promoter, there is also likely chromatin effect due to the position of insertion in the genome etc. Nevertheless, PSAD promoters are commonly used in *Chlamydomonas* for complementation purposes. Here are a few examples (<https://pmc.ncbi.nlm.nih.gov/articles/PMC10022639/>; [10.1093/emboj/cdg591](https://doi.org/10.1093/emboj/cdg591); [10.1111/j.1365-313X.2006.02870.x](https://doi.org/10.1111/j.1365-313X.2006.02870.x); [10.1093/jxb/erx343](https://doi.org/10.1093/jxb/erx343); [10.1038/s41598-019-39506-6](https://doi.org/10.1038/s41598-019-39506-6)) and there are many more. That said, we believe that in our case the use of a strong promoter did not give false-positive results, since the *lci20* phenotype was rescued and no additional phenotype was observed in the complemented lines.

3. L171-172: The study here mentioned the photorespiratory (21% O₂) or non-photorespiratory (2% O₂) conditions for the first time. The experimental conditions used should be validated. Since the ratio of (O₂/CO₂) is still very high under 2%O₂/VL-CO₂ condition, the non-photorespiratory conditions used here may not be convincing.

R: These conditions are commonly used in higher plants as non-photorespiratory conditions (doi.org/10.1038/s41467-023-42648-x; doi.org/10.1111/ppl.12146). Moreover, the absence of glycolate excretion in all strains, including *gyd* mutants, which cannot metabolize glycolate and therefore excrete it, clearly shows that these conditions are non-photorespiratory.

4. Line177-180 “Note that similar growth was observed when wild-type cells were exposed to VL-CO₂ either under photorespiratory conditions (21% O₂) or non-photorespiratory conditions (2% O₂), indicating that photorespiration is not required for CCM induction”. To confirm this, liquid culture results are required and are more convincing than solid plate culture by measuring the growth performance of the strains.

R: We have now performed growth analysis of wild-type *Chlamydomonas* in liquid cultures under VL-CO₂ at 21% and 2% O₂ and obtained similar growth rate at both O₂ concentrations. This new result is now added in the revised version as **Supplementary Fig. 5**.

5. Line190-192: “We conclude that *lci20* is defective in the photorespiratory glycolate metabolism, which is triggered when *Chlamydomonas* cells are transferred from H-CO₂ to VL-CO₂.” What about the phenotype of the regular *lci* mutant? Because of the high ratio (O₂/CO₂), glycolate excretion could also be induced in the regular *lci* mutant. Therefore, it's necessary to include the regular *lci* mutant as a control strain to demonstrate the function of LCI20.

R: It is not clear what the reviewer means here by “the regular *lci* mutant”. Actually a few other *lci* mutants have been described in the literature, which are affected in different CCM components, including for instance LCI11 (BST1-3), LCI1, LCI5 (EPYC1), LCIA, LCIB, LCIC,... The characterization of these mutants gave very different phenotypes. We agree that glycolate excretion may be triggered in CCM mutants, as we have shown for *cia5*, *cah3* and *bsti* mutants which we used as controls (see our answer to the major concern 2). The phenotypical difference between CCM mutants and *lci20* is the growth rescue occurring under VL-CO₂ at 2% O₂ (non-photorespiratory condition) in *lci20*. CCM mutants are unable to grow under VL-CO₂ regardless of O₂ levels as we have shown in **Supplementary Fig. 6**. Regarding glycolate excretion, we have shown that CCM mutants and *lci20* excrete large amounts of glycolate under photorespiratory conditions at VL-CO₂ 21% O₂ (**Fig. 2b; Supplementary Fig. 6**).

6. Line201-206: In general, the quality of the Western blot data is not convincing to support the hypothesis stated in the text. Upon careful examination, GCSP and HPR1 share the similar profiles to that of GYD1 under 21% O₂ or even under 2% O₂ condition when comparing the *lci20* and WT strain. To validate the results, multiple replicates of Western blot should be performed, and the quantification of the abundance profile of the target protein could be done using Image J software.

R: We have now performed additional western blot analysis and quantification from three biological replicates by using ImageJ software. These new data, supplied in **Supplementary Fig. 7 and 8** have been used to update Fig. 3. In our previous data in Fig. 3, GYD1 was less abundant in the *lci20* under photorespiratory condition. The new data showed the same exact phenotype (less GYD1 in *lci20*) and the difference between *lci20* compared to CC4533 is statistically significant. This decrease in GYD1 abundance in *lci20* was fully rescued in the complemented line. The *lci20* showed similar abundance of HPR1 and GCSP as in the wild-type CC4533 and the complemented line C1.

7. Line221-223: “However, the growth of both *gyd1* mutants was similar to the control wild-type strain after a transition from H-CO₂ to VL-CO₂ regardless of O₂ levels (Fig. 4d)”. To validate the results, growth data from the liquid culture is required and the growth profile on the solid plate is not sufficient to draw a conclusion.

R: Although in many studies algal growth is generally monitored on solid media by performing spot tests, we have now conducted additional experiments in liquid cultures in order to strengthen our conclusion. For this purpose, we have used instrumented photobioreactors operated as turbidostats and compared the growth properties of the WT and a *gyd1* mutant under different CO₂ regimes (HC, LC and VLC). In these experiments, biomass concentration was maintained at a constant level by constant addition of fresh medium (at a constant culture volume) and growth was assessed as the culture dilution rate. In parallel, dissolved CO₂ concentration available to algae was measured by using MIMS in order to determine that actual dissolved CO₂ levels correspond to the CO₂ levels corresponding to LC to VLC. Doing so, and in agreement to the spot test experiments, we could not observe any growth difference between the WT and the *gyd1-1* mutant under the different CO₂ regimes tested (**Supplementary Fig. 9**).

8. Line232-245: Section “Photosynthesis of *lci20* is affected under photorespiratory conditions” could be moved to the supplemental materials, as several studies have already reported the impairment of photosynthesis by photorespiration defects.

R: We do not share the reviewer’s view, since in contrast to C3 plants, very few photorespiratory mutants have been characterized in algae to date. In C3 plants, most photorespiratory mutants show a strong growth phenotype under photorespiratory conditions (air CO₂ levels), growth inhibition resulting from an accumulation of non-metabolized photorespiratory intermediates. In microalgae, the situation is not that clear. As mentioned above, a *Chlamydomonas glycolate dehydrogenase* mutant was isolated as a CO₂-requiring mutant, but the CO₂-requiring phenotype was not confirmed or characterized further (Nakamura et al. 2005 Can J Bot 83, 820). Moreover, the two independent *gyd* mutants that have been characterized in our study do not show any growth phenotype under low CO₂ (Fig. 4 and Supplemental Fig. 9). The only photorespiratory mutant showing a defect in growth under photorespiratory conditions is a mutant affected in phospho-glycolate phosphatase, the accumulation of phospho-glycolate being extremely harmful for cells (Suzuki et al. 1990 doi: 10.1104/pp.93.1.231). Therefore, the defect of photosynthesis observed in *lci20* under photorespiratory conditions is a novel phenotype in algae.

Reviewer #3 (Remarks to the Author):

This paper reports a large effort to better understand photorespiration in organisms that have CO₂ concentrating mechanisms, specifically *Chlamydomonas*. The conclusions are well supported and are a significant step forward. The methods are described in good detail. All data are provided in supplementary files. I don’t have any substantive concerns but can point out some trivial things for the authors’ consideration.

1. It took me some time to work out the name of the gene and protein. Because of the similarity of capital I, I and lower case l, I was confused for a while I figured out that it was “low (L) inorganic carbon (Ci, here as CI) 20, LCI20. But is it Ci or specifically CO₂? Help here would save readers time.

R: We have now made this clear as followed: Low-CO₂ Inducible 20 (*LCI20*) gene.

2. The paragraph on pages five and six is very long. Can it be broken up? Perhaps here “membrane26. The contribution” And or transporters43. By another way

R: Thanks for the comment, we have now split the paragraph in two parts.

3. Line 196, “Over excretion” sounds funny to me. How about “and large (or significant) glycolate excretion.”

R: We have used “large glycolate excretion” instead of “over excretion”.

4. Line 345 “decarboxylation of two molecules of glycine ...” gives the wrong impression. How about “conversion of two molecules of glycine to one serine...”

R: We have replaced “decarboxylation” into “conversion”.

5. Line 347 this enzyme requires ATP and so it is a synthetase, synthases do not use ATP

R: Thanks for pointing this out. It has been rectified.

6. Line 416 Glycolic acid was separated

R: Thanks for pointing this out. Now we have corrected the mistake.

7. 438 and elsewhere. I suggest italicizing all variables (as has been done for the different fluorescence parameters. That means q_L should also be italicized. Also, so many of the qualifiers - m' and s of F_m' and F_s - should be subscripted. Perhaps eventually the field will eliminate these rules but for now I suggest sticking with them.

R: Thanks for pointing this out. All is in italic now.

8. The references have a lot of issues with species names not italicized, capitalization of titles vary, subscripting is often absent. If the journal is going to reset the references from a database that will fix these then there is no concern but if the manuscript will be used to get these things right then some work needs to be done.

R: Thanks for pointing this out and it has been fixed.

9. Supplementary Fig 1a. The expression at end of the dark period increases from 20% to 100 % between the end of the night on the right side to the beginning of day at the left side.

R: We thank the reviewer for pointing this out. The data has now been updated, and the mistake has been removed (see new supplemental Fig.1). All data are from Zones et al.
(doi: 10.1105/tpc.15.00498)

Reviewer #1 (Remarks to the Author):

Mostly suggested some language clarification.

All of those have been addressed in the revised manuscript with track changes:
L72, L76, L86-87, L194, L247, L255, L2561, L303, L317, L386-387 and L395.